Published at the ICLR 2022 workshop on Objects, Structure and Causality

# FACTORIZED WORLD MODELS FOR LEARNING CAUSAL RELATIONSHIPS

**Artem Zholus, Yaroslav Ivchenkov**
Moscow Institute of Physics and Technology
Moscow, Russia
{zholus.aa,ivchenkov.yap}@phystech.edu

**Aleksandr Panov**
AIRI, MIPT
Moscow, Russia
panov.ai@mipt.ru

## ABSTRACT

World models serve as a powerful framework for model-based reinforcement learning, and they can greatly benefit from the shared structure of the world environments. However, learning the high-level causal influence of objects on each other remains a challenge. In this work, we propose CEMA, a structured world model with factorized latent state capable of modeling sparse interaction. This is possible due to a separate state and dynamics of three components: the actor, the object of manipulation, the latent influence factor between these two states. In multitask setting, we analyze the mutual information of the hierarchical latent states to show how the model can represent sparse updates and directly model the causal influence of the robot on the object.

## 1 INTRODUCTION

Reinforcement learning (RL) has recently benefited extensively from structural inductive biases. Assuming block structure of MDP (Du et al., 2021) or graph-structured objects interaction model (Veerapaneni et al., 2019) one can improve generalization in RL (Kirk et al., 2021). Other works bring structure into the world model of the agent to facilitate generalization, sample-efficiency, or casual modeling (Wang et al., 2020; Hallak et al., 2015). The latter is especially promising as world models (Ha & Schmidhuber, 2018) can tremendously benefit from the reuse of the learned or encoded structure of the world.

In this work, we present CEMA (Cause-Effect Modeling Agent), a world model that learns the correct cause-effect relationship between the actor and the object of manipulation. We show how the model can learn latent influence from the actor state on the object state. We hypothesize that the state of the manipulator should be modeled separately from the state of the object of manipulation. Each of them should have separate latent dynamics where actions control the former, and the latter is controlled by an abstract influence factor determining how the object state should be updated in response to the manipulator state update. This latent factor can be seen as a high-level action that directly controls the object state, the idea similar to (Eysenbach et al., 2021). We build the world model by employing these latent variables in a hierarchical fashion, and we show that prior distribution models the forward cause-effect relationship while the approximate posterior models the inverse process of reducing the effect to cause. In our experiments, we fix the setup to a known common cause case, so we do not need to discover it as it is fixed and known.

## 2 RELATED WORK

Model-based RL algorithms can significantly benefit from having a structured world model, especially in a visual control setting. PlaNet (Hafner et al., 2018) and Dreamer (Hafner et al., 2019) train a recurrent world model, separating deterministic and stochastic parts of environment dynamics. However, RPC (Eysenbach et al., 2021) argues that robust and generalizable behaviors also depend on compression in representation learning. Following the Information Bottleneck principle (Tishby et al., 2000; Alemi et al.,

2016), they build a compressed representation and restrict a policy to follow predictable paths. We follow a similar principle, combining it with a causal-based, object-centric world model.

The compound causal structure is necessary for modeling good object interactions. C-SWM (Kipf et al., 2020) adopts graph neural network for object-level reasoning, allowing for flexible relationship modeling. However, there is no explicitly stated causality between objects. OP3 (Veerapaneni et al., 2019) operates on entities rather than objects to simulate laws of the environment, equally valid for every object in it. While they achieve impressive generalization in combinatorially hard problems, modeling asymmetric or sparse causal relationships is not possible. In GATSBI (Min et al., 2021) authors model agent, background, and objects differently. The influence of entities on each other is designed as GNN, taking into account their different nature. Whereas this approach is suitable for scenes with numerous simple objects, it may lack the expressiveness for the case of complex actor-object interaction. A number of works on representation learning in object-centric context (Greff et al., 2019; Locatello et al., 2020; Engelcke et al., 2020) also use hierarchical latents to introduce a structure into the scene. However, they do not explore modeling dynamics in these terms.

## 3 BACKGROUND

### 3.1 REINFORCEMENT LEARNING

We formalize the visual control problem as a partially observable Markov Decision Process $\mathcal{M} = \langle \mathcal{S}, \mathcal{A}, \mathcal{O}, P, R, \gamma \rangle$, where $\mathcal{S}$ is the set of states, $\mathcal{A}$ is the set of actions, and $\mathcal{O}$ is the set of observations. The state dynamics follow the conditional transition distribution $P(s' \mid s, a)$, but the agent only has access to samples from the observation distribution $o \sim p(o \mid s)$, $o \in \mathcal{O}$. The agent is defined as a policy $\pi(a_t \mid o_{\leq t}, a_{<t})$. It interacts with a partially observable environment by taking actions on the environment and getting a next observation. We write $o_t, r_t \sim p(o_t, r_t \mid o_{<t}, a_{<t})$ as a shorthand for $s_t \sim P(s_t \mid s_{t-1}, a_{t-1})$, $o_t = O(s_t, a_{t-1})$, $r_t = R(s_t, a_{t-1})$. The goal of the agent is to maximize its expected discounted sum of rewards $\mathcal{R}(\pi, \mathcal{M}) = \mathbb{E}_\pi \sum_t \gamma^t r_t$. We also adopt notion of Context MDP (CMDP) introduced in (Kirk et al., 2021). A context is a latent variable that determines the task in CMDP. Each context defines the states that could be visited during an episode such as a level in level-based environments or the parametric environment configuration in environments that are parameterized by vector. Context variable allows to factorize the initial state distribution $p(s) = p(s' \mid c)p(c)$, where $s = (s', c)$. The observation is then sampled from the emission distribution, i.e., $o \sim p(o \mid s')$.

### 3.2 MODEL-BASED REINFORCEMENT LEARNING VIA WORLD MODELS

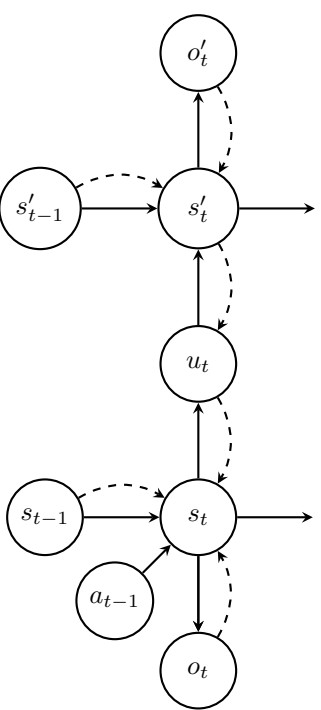

Figure 1: Graphical model for the factorized agent. Solid lines indicate generative distributions; dashed lines indicate inference distribution.

World models explicitly learn environment dynamics to generate novel virtual experience (Sutton, 1991; Ha & Schmidhuber, 2018). When observations are non-Markovian, e.g., in visual control, this can be achieved only by incorporating latent states. For latent dynamics learning, we use the RSSM (Hafner et al., 2018), which learns the dynamics by building a Markovian latent state for each timestep $s_t$ given previous action $a_{t-1}$ by autoencoding observations $o_t$ and rewards $r_t$, which are non-Markovian. Dreamer's world model consists of a representation model, or an encoder $q(s_t \mid s_{t-1}, a_{t-1}, o_t)$, a transition model $p(s_t \mid s_{t-1}, a_{t-1})$, an observation model $p(o_t \mid s_t)$, and a reward model $p(r_t \mid s_t)$. The representation and transition models share common parameters in the RSSM network (Hafner et al., 2018). The world model is trained to maximize the variational lower bound on the likelihood (ELBO) of

the observed trajectory conditioned on the actions $\mathbb{E}_p \log p(o_{1:T}, r_{1:T} \mid a_{1:T})$. This is done by incorporating an approximate posterior model $q(s_t \mid s_{t-1}, a_{t-1}, o_t)$, which is known as a representation model. This model acts as a proposal distribution for states $s_t$.

## 4 LATENT STATE FACTORIZATION

In this section, we describe a latent factorization we use in our method. In many cases, the scene components of the visual control problem can be divided into two groups based on their functional role: *acting* and *objective*. The first one represents everything that can be controlled directly by an agent, e.g., a robotic arm, as in our case. The latter stands for everything that controls what the agent can interact with.

We model each group using hierarchical latent variables with an additional latent influence factor. To provide correct signal for modeling both parts, the input image $\mathbf{o}_t$ is separated into corresponding parts via a segmentation mask: the robot image $o_t$ and the object image $o_t'$. The generative part of the model explicitly expresses the direct cause-effect relationships between the actor and the object, resulting in a bottom-up imagination process. Since the actor is assumed to be directly controlled by the MDP dynamics, we model its updates using robot state $s_t$ with latent transition model $p(s_t \mid s_{t-1}, a_{t-1})$ and observation model $p(o_t \mid s_t)$. The objective latent states $s_t'$, on the other hand, are not controlled directly via actions $a_t$. Instead, their control is mediated by the influence of the physical parameters of the actor on the object at each point in time. We express this collateral control using a latent influence factor $u_t$ that abstracts how the objective state should be changed to reflect the observational update. Since the actor state does not contain the information about the current task, we condition the influence factor on the MDP context vector $c$ in addition to the robot state $s_t$, which results in an influence model of the form $p(u_t \mid s_t, c)$. For the objective part, we use the prior distribution in the form of a Hidden Markov model (HMM) $p(s_t' \mid s_{t-1}', u_t)$ to model influenced objective updates and the objective observation model $p(o_t' \mid s_t')$. The posterior model inverts causal relationship and infers states in a top-down fashion. This hierarchical structure is motivated by the causal effects that the agent should model. We hypothesize that such architecture would allow the model to propagate the information from the cause, i.e., the robot, to an effect, i.e., the object movement.

The object posterior model $q(s_t' \mid o_t')$ infers the actual updated state solely from the updated observation. This choice was made to account for sparse objective dynamics as the recurrent stochastic model would pass the previous state as is and not account for sparsity. The inferred objective state is then used to produce the influence factor $u_t$ via the inverse influence distribution $q(u_t \mid s_t')$. This distribution reverses the causal process of influencing $s_t$ on $s_t'$ and produces a sample of $u_t$ that could lead to the actual updated object state $s_t'$. Using the embedding of the robot observation $o_t$, the model infers an updated robot state $s_t$ given the inverse influence factor $u_t$ and the previous state-action pair $s_{t-1}, a_{t-1}$ resulting in the robot posterior $q(s_t \mid s_{t-1}, a_{t-1}, u_t, o_t)$.

The world model optimizes the following lower bound:

$$\mathcal{L}(\theta) \geqslant \mathbb{E}_{s_t, u_t, s_t' \sim q} \log p(o_t', o_t \mid s_t', s_t) + \beta_S \mathcal{L}_S(\theta) + \beta_O \mathcal{L}_O(\theta) + \beta_U \mathcal{L}_U(\theta) \tag{1}$$

$$\mathcal{L}_S(\theta) = \mathrm{KL}(q(s_t \mid s_{t-1}, a_{t-1}, u_t, o_t) \| p(s_t \mid s_{t-1}, a_{t-1})) \tag{2}$$

$$\mathcal{L}_U(\theta) = \mathrm{KL}(q(u_t \mid s_{t-1}') \| p(u_t \mid s_t, c)) \tag{3}$$

$$\mathcal{L}_O(\theta) = \mathrm{KL}(q(s_t' \mid o_t') \| p(s_t' \mid s_{t-1}', u_t)) \tag{4}$$

We implement the acting prior and posterior dynamics via the RSSM model (Hafner et al., 2018). For the objective part, instead, we implement it using the following architecture:

$$h_{t+1}, f_{t+1} = \mathrm{GRU}(h_t, [s_t', u_{t+1}]), \quad \tilde{f}_{t+1} = \mathrm{MLP}(e_{t+1}) \tag{5}$$

$$p(s_{t+1}' \mid s_t', u_{t+1}) = \mathrm{Dist}(f_{t+1}), \quad q(s_{t+1}' \mid o_{t+1}') = \mathrm{Dist}(\tilde{f}_{t+1}) \tag{6}$$

Here, $e_{t+1}$ is a CNN embedding of the observation, and $Dist$ is any distribution such as Normal, Categorical, or Bernoulli.

The nature of the process dictates the structure of the objective models we represent with these distributions. RSSM is an appropriate choice for the controlled processes where there is a direct causal factor, such as $a_{t-1}$ for the actor modeling. Object state, however, could be changed quite rarely and is instead updated by after-effects of $a_{t-1}$. Therefore, we infer the latent object state only from its current observation $o'_t$, while the influence factor $u_t$ acts as a proxy of actor history that led to $s'_t$. The objective model takes form similar to the one in the RPC agent (Eysenbach et al., 2021). Note also, that an inverse influence factor affects the inferred robot dynamics. This factor explains derailments of robot state caused by, e.g., bumping object.

Figure 2: **Top row:** Examples of robot observations for different drawer angles. **Bottom row:** The decomposed input for the model, left image shows masked robot, right image shows masked drawer without robot occlusion. Images rendered at higher resolution for clarity.

## 5 EXPERIMENTS

**Rotated Drawer World.** To test how the model can infer the correct causal relationship between the robot and the object of control, we built a custom robotic visual control environment based on MuJoCo (Todorov et al., 2012) and MetaWorld (Yu et al., 2019). Our environment represents a Sawyer robot that stands before the table with a green drawer. The robot's goal is to open the drawer while the position of the drawer can vary between tasks. We parameterize the robot's task by a polar coordinate of the drawer. Namely, we fix the radius to make the task effectively dependent only on a polar angle, making the context set $\mathcal{C} = [0, 360) \subset \mathbb{R}$. Examples of the image observations are in Figure 2. A more specific description of the environment is provided in Appendix A. The motivation to use this setup is as follows. First, each task alone is simple for a visual RL algorithm to solve quickly. Second, the tasks share much common structure with only one variation parameter. Third, the topdown camera allows observing the drawer from the same angle for each task leaving the need of understanding complicated 3D object variations. These facts combined lead to the testbed that promotes fast learning of similar tasks and that is aimed at the reuse and understanding of the information extracted from those tasks.

**Baselines.** As a baseline, we train the Dreamer agent on the proposed set of tasks. We leave all hyperparameters unchanged compared to the original algorithm (Hafner et al., 2019). The model is trained for a million steps on tasks with contexts sampled from the uniform distribution over the training contexts set.

**Results.** The CEMA agent shows comparable performance with Dreamer in terms of final reward. Both models converge to approximately the same value of the episode return ($> 3500$). The convergence plot is present in Appendix B. The return above 2000 corresponds to a robot correctly approaching the drawer; the 3000 value of the return corresponds to a correct opening of the drawer. The difference between the returns above 3000 is the speed at which the robot opens the drawer. We found the non-recurrent object layer posterior $q(s'_t \mid o'_t)$ to be performing much better compared to a recurrent variant. This is probably due to the fact that the object updates are sparse and are performed for a short period of time under certain conditions (when the robot interacts with the object). Also, the inverse cause nature of the posterior distribution showed to be contributing to the final performance.

As the main contribution in this work, we show that the model reflects cause-effect relationship between the robot and the object of control. We found that this fact can qualitatively be expressed by studying mutual information (MI) between the input and output of each distribution. For example, for the influence model prior, we calculate $\mathbb{E}_c I(u_t, s_t \mid t, c) = \mathbb{E}_{c,s_t} \text{KL}(p(u_t \mid s_t, c) \| p(u_t \mid c)) \leqslant \mathbb{E}_{c,s_t} \text{KL}(p(u_t \mid s_t, c) \| \sum_{\hat{t}} p(u_t \mid s_{\hat{t}}, c))$. This is a variant of InfoNCE (van den Oord et al., 2019) estimator. That is, we compare the log-probability of the current sample $u_t \sim p(u_t \mid s_t, c)$ and the log of the average probability of this sample where average is taken over all timesteps within the contiguous batch. This forms an

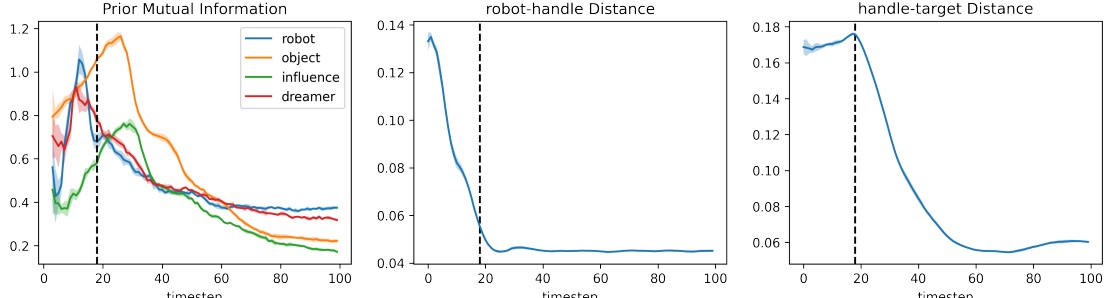

Figure 3: Results of the Mutual Information analysis. **Left:** per-timestep estimates of the mutual information of prior distributions for three latent layers of the model. **Middle:** distance between the robot end-effector and the drawer handle. **Right:** distance between the drawer handle and its target position.

upper bound on true mutual information. In Figure 3 we show the comparison of $\mathbb{E}_c I(u_t, s_t \mid t, c)$, $\mathbb{E}_c I(s'_t, (s'_{t-1}, u_t) \mid t, c)$, $\mathbb{E}_c I(s_t, (s_{t-1}, a_{t-1}) \mid t, c)$, i.e., we do not average over timesteps. The metric growth in the beginning of each plot cannot be explained by the burn-in effects of the neural network distribution because we calculate this metric in a bootstrapping fashion. This means, for each timestep, the metric value is obtained given a varying number of present previous timesteps (as each batch contains a short contiguous chunk of the episode), so the burn-in effect would result in a much higher variance. The mutual information metrics show that the prior part of the world model flows the activation of prior distributions in a bottom-up fashion corresponding to the direct influence of cause on effect. The robot state MI is activated during the period of the active robot movement, as shown in the middle of Figure 3. This, in turn, pumps up the MI of the influence latent and the object state. The vertical dashed line indicates the event when the robot starts moving the drawer. The object latent mutual information is maximized at some moment after the drawer starts to move. We hypothesize that the described behavior can be explained by reactive hierarchical dependencies between the robot state, the influence factor, and the object state that reflects a causal relationship between them. In other words, each individual MI value does not express the nature of causal dependency, but a combination of them, their co-evolution in time plays as an indicator of correct causal dependencies. This is also supported by the fact that the prior distributions reflect this property so they produce the excitement given only sequences of the robot actions as a common cause. In our experiments, we work with multitask environments parameterized by task context $c$. This is done for two reasons: to additionally ensure that the metric behavior is reproducible between the tasks and to ground the work on generalization in task learning in RL, which we aim at for the future work on this model.

## 6  DISCUSSION AND CONCLUSION

In this work, we presented CEMA, a structured world model that is capable of multitask reinforcement learning via correct modeling of cause-effect relationships. The model maintains the state of the actuator, the influence of the actuator on the object, and the object state. Using a mutual information estimator, we showed that in proposed model the information flows from cause to effect state as long as the interaction happens. The main limitation of the model is that the cause and effect should be manually encoded into the architecture of this model, making it unable to discover causality. Also, we do not assume any internal structure of the space of objects, i.e., the whole set of objects is modeled via one latent. The proposed model shows the principal way how the generalization with a world model can be achieved: through structural inductive bias, the compression of the information, and restricting models to act on limited input. We plan to research these ideas in the future work on this model. Another prominent direction of improvement is eliminating the ground truth segmentation assumption. It could additionally reveal the inferred causal structure, for example, if an attention would be used so the model could learn which parts of image to attend to. This can be an additional sign of correct causal modeling.

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

## A DETAILS OF IMPLEMENTATION

**Rotated Drawer Environment** For all tasks, the drawer's handle is oriented to the coordinate center, which coincides with the initial position of the drawer's gripper. The virtual camera is placed straight above the coordinate center so all drawer positions can be viewed from the same angle in the observation. The action space is the same as in MetaWorld, a four-dimensional vector representing shift vector for the gripper and normalized torque applied to the gripper fingers. The observation is a $64 \times 64$ RGB image. Examples of the image observations are in Figure 2. The reward function is defined in the same way as in the DrawerOpen-v2 task of MetaWorld (Yu et al., 2019). Namely, the reward is normalized in the range $[0, 10]$ where maximal reward corresponds to the solved task. The length of each episode is equal to 500.

**Agent architecture.** We use the ground truth actor-object segmentation provided by Metaworld environment. Also, to provide the model with an unoccluded image of the object, we render an additional image with a transparent robot. This way, we naturally avoid an undesired behavior of the model via robot occlusion understanding. Actor RSSM implementation with all its parameters is identical to Dreamer. All distributions in the model are assumed to be Gaussian. Both prior and posterior distributions of the influence factor are represented by a single-layered MLP with 200 units. The object layer has the same imagination mechanism as in Dreamer, but instead of actions, it is conditioned on the influence factor. Also, we use the deterministic part of the object state only in imagination and disregard it anywhere else since we assume that most of the information is contained in the current image. Instead of predicting the full image, separate decoders for object and actor images are used. Changes were made to the training process of the object layer to account for the specifics of the object layer. In particular, the free KL divergence gap between object prior and posterior originally introduced in the Dreamer algorithm, was removed, and the corresponding loss weight was increased by three times. We justify these changes with three following reasons: to account for low stochasticity of object appearance, to employ ideas from RPC (Eysenbach et al., 2021), and to additionally push prior distribution towards posterior, since learning influence vector through completely stochastic object state is difficult for the model.

## B RL AGENTS PERFORMANCE

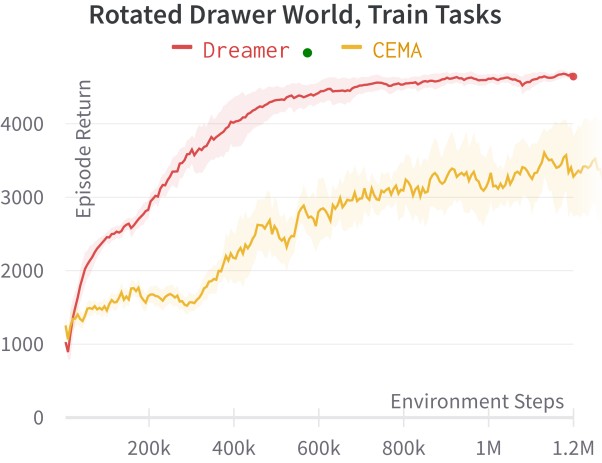

Figure 4: The performance plot of the Dreamer and CEMA agents. Note that the reward above 3000 corresponds to a solved task, with values above it reflecting the speed of task solving.

## C  MUTUAL INFORMATION UPPER BOUND DERIVATION

For example, consider an influence factor prior distribution $p(u_t \mid s_t, c)$:

$$
\begin{aligned}
\mathbb{E}_c I(u_t, s_t \mid c) &= \mathbb{E}_c \mathrm{KL}(p(u_t, s_t \mid c) \| p(u_t \mid c) p(s_t \mid c)) \\
&= \mathbb{E}_{c,s_t} \mathrm{KL}(p(u_t \mid s_t, c) \| p(u_t \mid c)) \\
&= \mathbb{E}_{c,s_t,u_t} \log p(u_t \mid s_t, c) - \log p(u_t \mid c) + \log \sum_{\hat{t}} p(u_t \mid s_{\hat{t}}, c) - \log \sum_{\hat{t}} p(u_t \mid s_{\hat{t}}, c) \\
&= \mathbb{E}_{s_t,c} \mathrm{KL}(p(u_t \mid s_t, c) \| \sum_{\hat{t}} p(u_t \mid s_{\hat{t}}, c)) - \mathbb{E}_{c,s_t,u_t} \log p(u_t \mid c) - \log \sum_{\hat{t}} p(u_t \mid s_{\hat{t}}, c) \\
&= \mathbb{E}_{s_t,c} \mathrm{KL}(p(u_t \mid s_t, c) \| \sum_{\hat{t}} p(u_t \mid s_{\hat{t}}, c)) - \mathbb{E}_c \mathrm{KL}(p(u_t \mid c) \| \sum_{\hat{t}} p(u_t \mid s_{\hat{t}}, c)) \\
&\leqslant \mathbb{E}_{s_t,c} \mathrm{KL}(p(u_t \mid s_t, c) \| \sum_{\hat{t}} p(u_t \mid s_{\hat{t}}, c))
\end{aligned}
$$

This way we recovered an upper bound for the task-averaged mutual information. For other distributions, all latent factors should be explicitly conditioned on the task context $c$. Effectively, this task context may be omitted in the final formulas given the structure of our graphical model. This is an upper bound since we approximate the marginal distribution in the RHS of KL. Our estimator simply averages probability scores of a $u_t$ sample over all distributions in time. The bound has the form similar to InfoNCE (van den Oord et al., 2019) without a critic which is why it has low bias.

