# OpenReview forum: "Factorized World Models for Learning Causal Relationships"
_ICLR.cc/2022/Workshop/OSC — ICLR2022 OSC  Poster_

### Official Review · Reviewer_LZ65 · 2022-03-16

**Rating:** 3
**Confidence:** 3

**Review:**

The paper proposes a method for learning structured world models by capturing cause and effect relationships between actors and obects.
Pros of the paper
1. The paper is very well written, it is clear and easy to follow.
2. The paper looks at an very interesting problem of learning cause-effect relationships between actors and objects in a world model. This is a very promising direction of research, since learning cause and effect relationships could allow models to generalize much better, especially OOD.
3. The method is interesting and makes sense, thanks for the evaluating on the mutual information between states.

There are a few things that might be interesting to take into consideration in the next version.
1. I believe that this method assumes  certain structures in the cause-effect relationship. For example, the method assumes that there is a bipartite relationship between actors and objects only. For example, if the actor acted on object A, and later, the changes in object A effects object B. Im not sure the current model could capture this. However, this is very important for causal learning (as most of the time, the causal structure is not a bipartite graph).  The method also assumes that the object can not influence the actor, which may not always hold true in the real world. It might be nice to include a section that discusses about this.
2. I believe the method can not necessarily disentangle correlations in the data. It might be nice to include a section that discuss about this in the paper.
3. There is a very related paper called "Neural Production Systems" at NeurIPS 2021. The NPS method can similarly learn relationships between objects and rules. It might be nice to include a discussion about this paper.
4. Lastly, it might be interesting to see the method working on other tasks that involves more complicated transfers between training and test tasks.

Overall, the paper is very well written, the method is interesting and I would be very excited to see further experiments and results on this method.

---

### Official Review · Reviewer_VoxZ · 2022-03-19
**A novel combination of ideas from causality and model based reinforcement learning**

**Rating:** 2
**Confidence:** 2

**Review:**

I thank the authors for their submission, which I enjoyed reading. Overall I found the manuscript to present an interesting combination of ideas relating to causality and reinforcement learning and will therefore be a valuable addition to the workshop.

Detailed comments

The authors propose cause effect modeling agent (CEMA) - a generative network which is designed to infer cause-effect relationships between an actor and the object of manipulation within an RL setting. The proposed model is able to infer latent influence from the actor state on the object state.

The authors claim that for an agent to exhibit robust and generalizable behaviors, they must incorporate some degree of “compression” within their learned representation of their environment. In this work, they explore the use of latent state factorization to achieve this goal and propose a hierarchical latent variable model to model visual control problems. In particular, they propose to decompose an input image into a robot image component (actor) and an object image component (object that is acted upon). This separation is implemented via segmentation mask. In this way they explicitly parameterize the cause-effect relationship between actor and object.

Via a series of experiments, the authors demonstrate that CEMA (for which the agent follows a factorized generative model) is able to obtain comparable performance to Dreamer (Hafner et al., 2019), thus showing that it is possible to introduce structure knowledge (e.g., about the cause effect structure) into the agent.

The authors further claim their proposed model is able to infer cause-effect relationships by studying the mutual information between the inputs/outputs of each distribution. For example, the authors claim that “object latent mutual information is maximized at some moment after the drawer starts to move.” I must admit that I did not fully understand why changes in the mutual information between variables is indicative of inferred causal structure - could it not be possible that the MI between variables changes without causal associations? It’s also unclear how pre-imposed causal structure affects these quantities. Would it be possible add similar plots for e.g., Dreamer as a comparison ?

Finally, one experiment that would go a long way towards convincing me that cause-effect relationships had indeed been inferred would be to replace the mask segmentation which is used to separate input images into actor/object images with an attention layer. Then each component of the factorized agent would have access to the entire image and would need to learn which components of it to attend to. Simple attention maps or saliencies of the image could then demonstrate that the actor and object components where indeed focusing on the right aspects of the image. This would also demonstrate that the proposed method can potentially scale to scenarios when the segmentation of images is not available.

Typos/minor:

- Section 5: “In figure 4 we show the comparison of $I(u, (s,c)), ..$. Do the $u$, $s$ and $c$ variables require a time subscript here (i.e., $u_t$, etc) ?
- section 5: “The robot’s goal is to open the drawer while the position of the can vary between tasks” -> missing noun
- Figure 4: it would be helpful to make vertical line dashed/dotted to avoid color confusion (e.g., between the red and orange lines)

---

### Decision · Program_Chairs · 2022-03-21

Accept (Poster)